genetics

Hui, Indel, individual identification, ethno-origin, admixture

**Author for correspondence:**
Ruiming Shi
e-mail: shiruiming75@126.com

# Genetic affinity between Ningxia Hui and eastern Asian populations revealed by a set of InDel loci

Boyan Zhou[1,2], Shaoqing Wen[2], Huilin Sun[3], Hong Zhang[4] and Ruiming Shi[5]

[1]State Key Laboratory of Genetic Engineering and Institute of Biostatistics, School of Life Sciences, and [2]State Key Laboratory of Genetic Engineering and MOE Key Laboratory of Contemporary Anthropology, School of Life Sciences and Institutes of Biomedical Sciences, Fudan University, Shanghai 200438, People's Republic of China
[3]Department of Endocrinology, The First Affiliated Hospital of Guangdong Pharmaceutical University, Guangdong 510080, People's Republic of China
[4]The First Affiliated Hospital Health Center and School of Management, University of Science and Technology of China, Hefei 230026, People's Republic of China
[5]Department of Pediatrics, the First Affiliated Hospital of Xi'an Jiaotong University, Xi'an 710061, People's Republic of China

BZ, 0000-0001-7067-4578

According to historical records, ethnic Hui in China obtained substantial genetic components from western Eurasian populations during their Islamization. However, some scholars believed that the ancestry of Hui people were native Chinese populations. In this context, the formation of Hui is due to simple cultural diffusion rather than demic diffusion. In this study, we examined the forensic and population genetic application of the 30 InDel loci in Hui population from Ningxia Hui Autonomous Region, Northwest China. Genotype analysis of 129 unrelated individuals revealed that all loci were in the Hardy–Weinberg equilibrium in Ningxia Hui. Forensic indices calculated from genotypes demonstrated that this panel, Qiagen DIPplex® Investigator kit, was powerful enough to be used in individual identification but not in paternity cases. Through population genetic analysis, we found that Ningxia Hui received much more genetic contributions from East Asian populations than those from western Eurasian populations. Finally, we statistically identified the admixture signal of eastern and western Eurasians, although the latter is weak, in Ningxia Hui via the three-population test. All this evidence suggested that the formation of Ningxia Hui was mainly attributed to the cultural transformation of local Chinese residents with minor gene flow from western Eurasian populations.

# 1. Introduction

China is a unified multi-ethnic country with 56 officially recognized ethnic groups. Having a sizeable population of approximately 10.6 million, Hui is the largest one among 10 Muslim ethnic groups in China according to the 2010 national census (http://www.stats.gov.cn/tjsj/pcsj/rkpc/6rp/indexce.htm). The term 'Hui' was derived from Mandarin word 'Huihui' which referred to Central Asians, Persians and Arabs residing in China in the age of the Yuan Dynasty (1271–1368). On the basis of Chinese historical records, Hui community originated from intermarriage between Han females and non-Chinese males [1,2] in approximately 120 BC. Most of these non-Chinese males were foreign merchants travelling from Central Asia along the Silk Road [3,4]. In this context, Hui is an admixed population of the East and the West. As a consequence, their western Asian genetic make-up can be easily observed.

However, compared with Uigur, a typical admixture of eastern and western Eurasians in both anthropometric and genetic traits [5], the origin of Hui has long been a controversial issue. Through the analysis of 15 autosomal short tandem repeats (STRs), Hui in Linxia, Gansu province, showed significant genetic homogeneity with East Asian populations [6]. This conclusion was also supported by the study of some other autosomal markers, such as human leukocyte antigen (HLA) class I polymorphisms [7]. In addition, some researches aimed to clarify the paternal genetic structure of Hui ethnic groups residing in different regions of China. Firstly, a high diversity of Hui's Y-chromosomal haplogroup was observed in northwest China [8]. Secondly, some studies with 15 or more Y-STRs revealed the close relationships between Hui and multiple ethnic groups in different districts [9–12]. However, the analysis of molecular variance (AMOVA) of eight Y-STRs suggested that the paternal origins of Hui are distinct from those of Han in Liaoning Province though close to eastern Asian groups [13].

Small insertion and deletion (InDel), ranging from 1 to 10 000 bp in human genomes [14], is a type of biallelic marker with a low mutation rate compared with STRs [15]. Due to the advantage of short amplicon size, InDel can be used for the analysis of degraded or ancient DNA samples [16]. Owing to the merits of no stutter peaks [17] and simple genotyping protocols, InDel is attracting more and more attention of the forensic community [18,19]. The Qiagen DIPplex® Investigator kit, which can multiplex 30 autosomal InDels and Amelogenin for forensic use, has become a widely used tool in population genetic analyses [20].

In this study, we assessed the forensic efficiency of this kit in a Hui population from the Ningxia Hui Autonomous Region. Combining the InDel data with that from other studies, we revealed the autosomal genetic relationships between Ningxia Hui and other populations. Although contributions have been made to the identification of admixture evidence in Hui, to the best of our knowledge, no unambiguous conclusions have been made in the literature. Using some population genetics analysis statistics, we identified an evident admixture signal of eastern and western Eurasians in this Hui population.

# 2. Material and methods

## 2.1. Sample collection, DNA extraction and genotyping

We collected bloodstain samples of 129 unrelated healthy Hui individuals in the Ningxia Hui Autonomous Region, China according to their identity cards. No kinship existed among the samples within at least three generations, and no migration events happened in their family history as declared. All of the donors had been adequately informed and signed the informed consent before sample collection. This study was approved by the ethics committee of Xi'an Jiaotong University Health Science Center and conducted in accordance with the human and ethical research principles of Xi'an Jiaotong University Health Science Center, China. The DNA was extracted using the Chelex® method (Solarbio, Beijing, China) and the genotyping of 30 InDels was carried out using the DIPplex Investigator reagent (Qiagen, Hilden, Germany), which followed established methods [20,21]. To perform genetic analysis of the Ningxia Hui population, we also collected 30 InDels data from 21 other populations (three Han populations [22–24], Yi [25], Xibe [20], South Korean [26], She [23], Tibetan [24], Kazak [24], Uigur [24], five European populations [18,27–29] and six Mexican populations [30]).

## 2.2. Data analysis

To assess the forensic efficiency of this panel, forensic parameters and allele frequencies were calculated using the modified powerstat (v. 1.2) spreadsheet (Promega, Madison, WI, USA). Pairwise $F_{ST}$s that

measure genetic distances between populations were computed using ARLEQUIN v. 3.5.2 [31] and visualized by multidimensional scaling (MDS) using SPSS v. 19 [32]. A projection principal component analysis (PCA) was used to investigate genetic affinities between Hui and other populations by smartpca program in EIGENSOFT 6.0.1 [33]. By setting the number of ancestral populations (K), the admixture analysis was performed using ADMIXTURE 1.23 [34] with default parameters. The three-population test is a formal test of admixture and can provide clear evidence of admixture. We carried out the three-population test to detect admixture signals in Ningxia Hui using qp3Pop program in ADMIXTOOLS (v. 410) [35]. All figures in this paper were redrawn using R statistical software v. 3.2.1 [36].

# 3. Results

## 3.1. Forensic analysis

Basic forensic indices and allele frequencies of the 30 InDel loci are summarized in table 1. The expected heterozygosity (He) that was calculated from allele frequencies ranged from 0.241 (HLD118) to 0.500 (HLD6 and HLD77). The observed heterozygosity (Ho) was in the range of 0.240–0.574 with the minimum at HLD39 locus and the maximum at HLD77 locus. No significant deviation from the expected value on any of the 30 InDel markers was found via the Hardy–Weinberg equilibrium test, with the minimal $p$-value being 0.102 (HLD77). The polymorphism information content (PIC) values ranged from 0.21 to 0.37. The power of exclusion (PE) ranged from 0.042 (HLD39) to 0.260 (HLD77) with a combined PE of 0.9940. The combined PE value was relatively low, suggesting that this panel can be used in combination with other markers, such as autosomal STRs, in paternity cases. The highest and lowest discrimination power (DP) was found at HLD6 (0.627) and HLD118 (0.396) respectively. Thus, the combined DP had a high enough value of 0.9999999999871, which provides a satisfactory level of discrimination for two randomly selected individuals in this group.

## 3.2. Genetic distances

To understand the genetic background of the Ningxia Hui population, we compiled obtained data with those from other 21 populations (three Han populations [22–24], Yi [25], Xibe [20], South Korean [26], She [23], Tibetan [24], Kazak [24], Uigur [24], five European populations [18,27–29] and six Mexican populations [30]). Pairwise $F_{ST}$s between these populations were calculated to show their genetic distances (electronic supplementary material, table S1). For this Hui population, the minimal $F_{ST}$ (0.00078) was observed for Han from Shanghai and the maximal value (0.21394) was observed for a group of Amerindian Mexicans. In general, the populations from East Asia, especially Han populations, had much closer relationships ($F_{ST} < 0.023$) with Hui than European populations ($0.055 < F_{ST} < 0.075$). As expected, Mexican populations had the most distant relationships ($F_{ST} > 0.13$) with Hui.

To visualize distances between these groups, we performed MDS analysis based upon linearized pairwise $F_{ST}$ values [37]. As shown in figure 1, the populations from East Asia mainly clustered at the bottom-right of the figure and the European populations were distributed on the top of the figure. Six Mexican populations were located at the bottom-left of the plots. Uigur and Kazak were in the central position, indicating an obvious admixture of eastern and western Eurasians, which was consistent with previous studies [5,38]. The Hui group marked with brilliant blue was on the border line of East Asian groups, which showed a distinction between Hui and typical mixed populations (Uigur and Kazak in this analysis).

## 3.3. PCA and admixture analysis

PCA was performed using individual InDel loci and was in agreement with the results above. The first and second components accounted for 13.62% and 4.76% of the total variance, respectively. Although not every individual could be assigned unambiguously to three main sources, namely, East Asians (Han, Yi, Xibe, Tibetan, South Korean and She), Europeans (Dane, Hungarian, Basque, Central Spanish and Uruguayan) and Mexicans, the PCA plots roughly fell into three parts (figure 2). The Central Asians (Uigur and Kazak) represented by green dots scattered in the middle of East Asians (yellow dots) and Europeans (blue dots). The Hui population (black dots), however, almost fell in the East Asians and did not show any tendency of admixture.

**Table 1.** Allele frequencies and forensic parameters for 30 InDel loci in Chinese Hui ethnic group in the Ningxia Hui Autonomous Region ($n = 129$). HLD, human locus deletion/insertion polymorphism; DIP−, frequency of short allele; DIP+, frequency of long allele; Ho, observed heterozygosity; He, expected heterozygosity; $p$, $p$-value for the Hardy–Weinberg equilibrium; PIC, polymorphic information content; PE, power of exclusion; DP, discrimination power; TPI, typical paternity index.

| HLD | rs# | DIP− | DIP+ | Ho | He | $p$ | PIC | PE | DP | TPI |
|---|---|---|---|---|---|---|---|---|---|---|
| 6 | 1 610 905 | 0.488 | 0.512 | 0.496 | 0.500 | 0.900 | 0.37 | 0.184 | 0.627 | 0.99 |
| 39 | 17 878 444 | 0.857 | 0.143 | 0.240 | 0.245 | 0.867 | 0.22 | 0.042 | 0.399 | 0.66 |
| 40 | 2 307 956 | 0.333 | 0.667 | 0.434 | 0.444 | 0.783 | 0.35 | 0.136 | 0.569 | 0.88 |
| 45 | 2 307 959 | 0.415 | 0.585 | 0.504 | 0.486 | 0.707 | 0.37 | 0.191 | 0.608 | 1.01 |
| 48 | 28 369 942 | 0.585 | 0.415 | 0.519 | 0.486 | 0.467 | 0.37 | 0.205 | 0.600 | 1.04 |
| 56 | 2 308 292 | 0.395 | 0.605 | 0.512 | 0.478 | 0.472 | 0.36 | 0.198 | 0.597 | 1.02 |
| 58 | 1 610 937 | 0.593 | 0.407 | 0.550 | 0.483 | 0.135 | 0.37 | 0.236 | 0.579 | 1.11 |
| 64 | 1 610 935 | 0.217 | 0.783 | 0.326 | 0.340 | 0.708 | 0.28 | 0.075 | 0.506 | 0.74 |
| 67 | 1 305 056 | 0.298 | 0.702 | 0.395 | 0.418 | 0.565 | 0.33 | 0.111 | 0.580 | 0.83 |
| 70 | 2 307 652 | 0.422 | 0.578 | 0.473 | 0.488 | 0.699 | 0.37 | 0.165 | 0.625 | 0.95 |
| 77 | 1 611 048 | 0.512 | 0.488 | 0.574 | 0.500 | 0.102 | 0.37 | 0.260 | 0.580 | 1.17 |
| 81 | 17 879 936 | 0.174 | 0.826 | 0.318 | 0.287 | 0.472 | 0.25 | 0.071 | 0.454 | 0.73 |
| 83 | 2 308 072 | 0.620 | 0.380 | 0.512 | 0.471 | 0.379 | 0.36 | 0.198 | 0.590 | 1.02 |
| 84 | 3 081 400 | 0.275 | 0.725 | 0.442 | 0.399 | 0.337 | 0.32 | 0.141 | 0.548 | 0.90 |
| 88 | 8 190 570 | 0.438 | 0.562 | 0.488 | 0.492 | 0.894 | 0.37 | 0.178 | 0.623 | 0.98 |
| 92 | 17 174 476 | 0.550 | 0.450 | 0.543 | 0.495 | 0.298 | 0.37 | 0.228 | 0.596 | 1.09 |
| 93 | 2 307 570 | 0.422 | 0.578 | 0.519 | 0.488 | 0.503 | 0.37 | 0.205 | 0.603 | 1.04 |
| 97 | 17 238 892 | 0.686 | 0.314 | 0.457 | 0.431 | 0.568 | 0.34 | 0.153 | 0.574 | 0.92 |
| 99 | 2 308 163 | 0.171 | 0.829 | 0.279 | 0.284 | 0.901 | 0.24 | 0.055 | 0.445 | 0.69 |
| 101 | 2 307 433 | 0.585 | 0.415 | 0.504 | 0.486 | 0.707 | 0.37 | 0.191 | 0.608 | 1.01 |
| 111 | 1 305 047 | 0.860 | 0.140 | 0.279 | 0.241 | 0.313 | 0.21 | 0.055 | 0.402 | 0.69 |

(Continued.)

**Table 1.** (Continued.)

| HLD | rs# | DIP− | DIP+ | Ho | He | $p$ | PIC | PE | DP | TPI |
|---|---|---|---|---|---|---|---|---|---|---|
| 114 | 2 307 581 | 0.694 | 0.306 | 0.395 | 0.425 | 0.474 | 0.33 | 0.111 | 0.586 | 0.83 |
| 118 | 16 438 | 0.860 | 0.140 | 0.248 | 0.241 | 0.853 | 0.21 | 0.044 | 0.396 | 0.66 |
| 122 | 8 178 524 | 0.733 | 0.267 | 0.395 | 0.391 | 0.963 | 0.32 | 0.111 | 0.553 | 0.83 |
| 124 | 6481 | 0.446 | 0.554 | 0.535 | 0.494 | 0.377 | 0.37 | 0.220 | 0.600 | 1.08 |
| 125 | 16 388 | 0.612 | 0.388 | 0.450 | 0.475 | 0.540 | 0.36 | 0.147 | 0.621 | 0.91 |
| 128 | 2 307 924 | 0.671 | 0.329 | 0.426 | 0.442 | 0.694 | 0.34 | 0.131 | 0.596 | 0.87 |
| 131 | 1 611 001 | 0.612 | 0.388 | 0.465 | 0.475 | 0.794 | 0.36 | 0.159 | 0.615 | 0.93 |
| 133 | 2 067 235 | 0.643 | 0.357 | 0.496 | 0.459 | 0.419 | 0.35 | 0.184 | 0.586 | 0.99 |
| 136 | 16 363 | 0.477 | 0.523 | 0.566 | 0.499 | 0.140 | 0.37 | 0.252 | 0.584 | 1.15 |

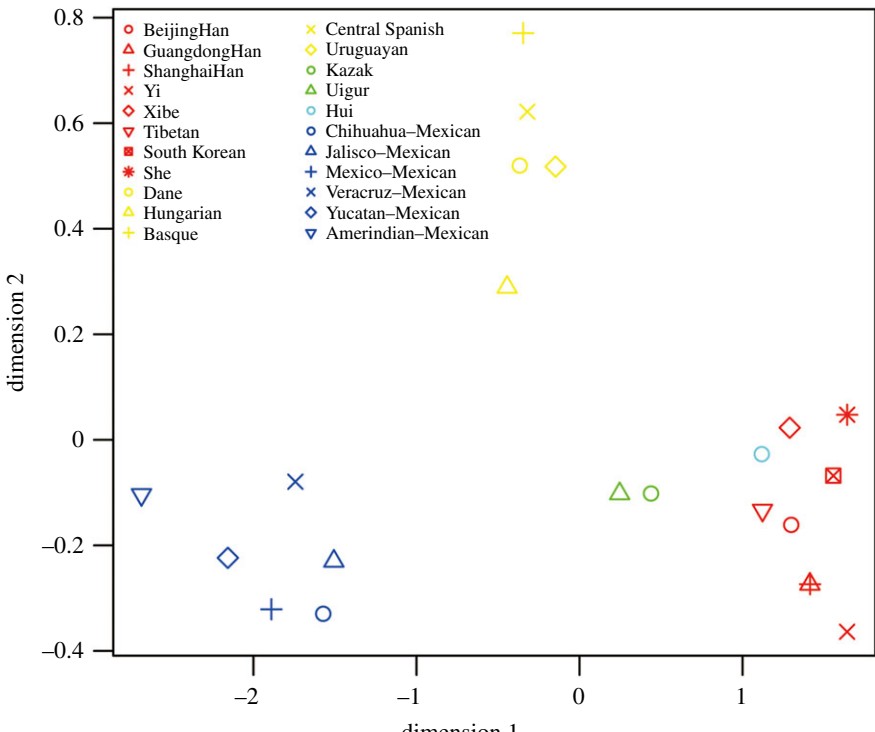

**Figure 1.** MDS plot of linearized pairwise $F_{ST}$ values. Brilliant blue, Hui; Red, East Asian; Yellow, European; Blue, Mexican; Green, Central Asian.

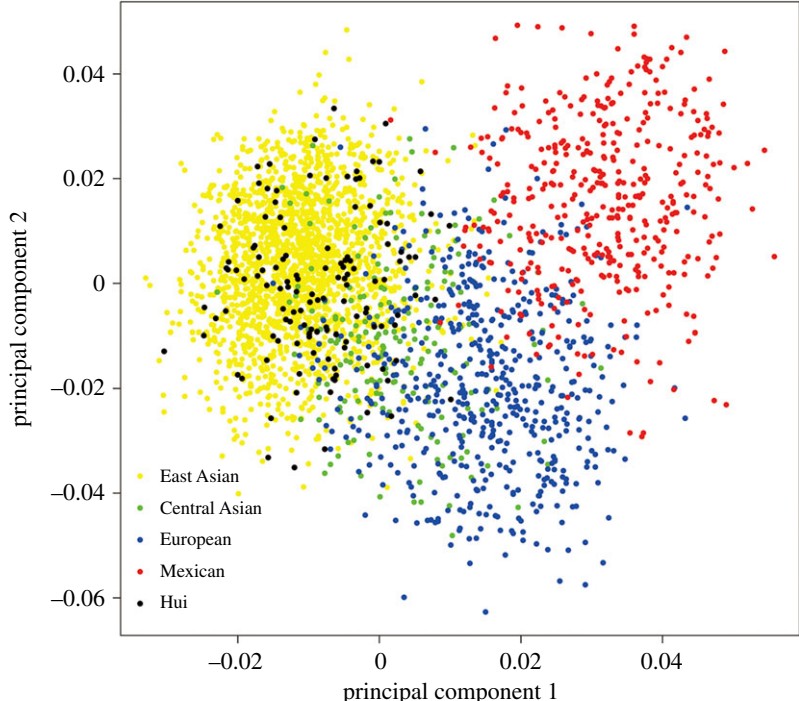

**Figure 2.** PCA plot at individual level. Black, Hui; Yellow, East Asian; Blue, European; Red, Mexican; Green, Central Asian.

We further conducted an admixture analysis on all individuals using ADMIXTURE [34] (figure 3). Because the number of loci is limited and these 30 InDels are not ideal ancestry-informative markers that exhibit substantially different frequencies between groups, colours in figure 3 should not be simply attributed to various sources of ancestries, such as Europeans, East Asians and Mexicans. Although the 30 InDel loci were not powerful enough to distinguish ancestries of these groups, they still offered circumstantial evidence of the genetic affinity between Hui and other East Asian populations. No matter

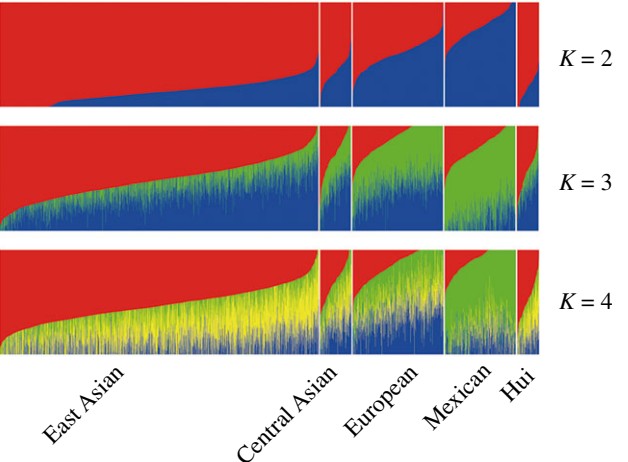

**Figure 3.** Admixture analysis of 22 populations with various group numbers ($K = 2$, 3, 4).

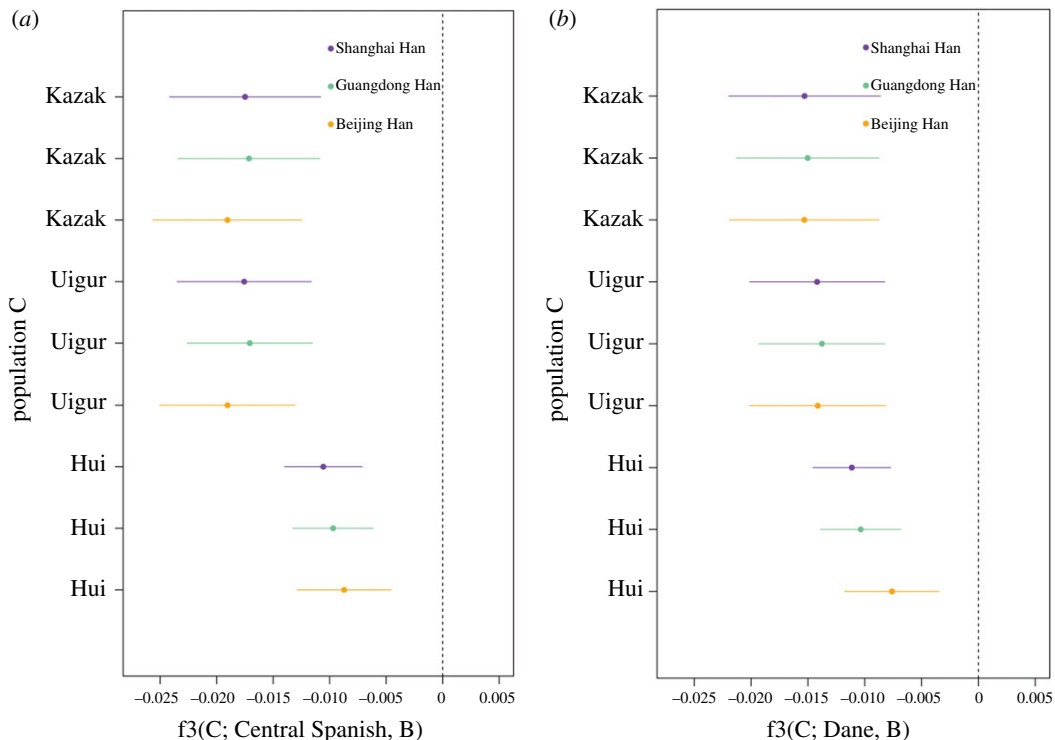

**Figure 4.** Three-population test of admixture. (*a*) f3(C; Central Spanish, B); (*b*) f3(C; Dane, B). C: Hui, Uigur and Kazak, B: Shanghai Han, Guangdong Han and Beijing Han. Horizontal lines represent standard errors.

what the value of $K$ (number of hypothetical ancestral populations) was, the pattern of Hui people resembled that of eastern Asians and was distinct from those of Central Asians and western Eurasians.

## 3.4. Three-population test

In the three-population, the so-called f3 statistic based on allele frequencies across populations can provide a clear evidence of admixture, and a negative value indicates a history of admixture [35]. Specifically, f3(C; A, B) < 0 is an evidence for population C having an ancestry from both A and B, and the admixture does not always result in a negative f3 when population C underwent a high degree of drift [35]. The resultant f3(Hui; Central Spanish, Han) were significantly negative with Z-scores < −2, which resembles f3(Uigur; Central Spanish, Han) and f3(Kazak; Central Spanish, Han) (figure 4*a* and electronic supplementary material, table S2). This result remained valid when

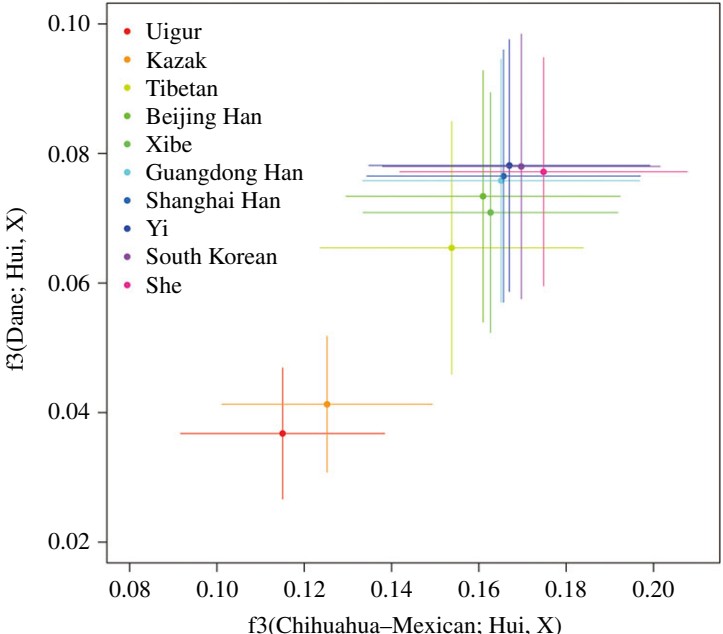

**Figure 5.** Three-population test of outgroup cases. X-axis: f3(Chihuahua-Mexican; Hui, X), Y-axis: f3(Dane; Hui, X), X: Asian populations. Horizontal and vertical lines represent standard errors.

Central Spanish was replaced with Dane (figure 4b). Therefore, we could reach a conclusion that western Eurasians had limited genetic contribution to Ningxia Hui, compared with their contributions to Uigur and Kazak.

In the outgroup case, the three-population test can also be employed to measure genetic affinities between populations [39]. For example, in f3(C; A, B), population C is an outgroup of A and B, and a larger f3 value means a closer relationship [40]. We set Danes and a Mexican population as outgroups and computed the f3(Danes or Mexicans; Hui, Asian groups) to evaluate affinities between Hui and other Asian populations. Even if taking into account the large confidence intervals, we still could find that Hui was much closer to East Asian groups than Uigur or Kazak (figure 5 and electronic supplementary material, table S3), which also indicated that the genetic contribution of western Eurasians to Hui was relatively weak.

# 4. Discussion

In this study, we tested the 30 InDels panel in the Hui population from Ningxia Hui Autonomous Region and found it highly efficient in the identification of unrelated individuals. However, this panel should be used in combination with other markers to achieve enough power in paternity cases. Since this panel was originally designed for individual identification and all loci were highly polymorphic in the majority of populations worldwide [18,24,26,30,41], its efficacy for ancestry inference might be limited as previously proved [20].

Nevertheless, this panel still could shed some light on the genetic relationships between populations [42]. Based on the analysis of $F_{ST}$s, we found that Ningxia Hui people were genetically closest to East Asian, especially Han groups. Despite limited samples and loci, our results were consistent with that calculated from the 1000 Genome Data [43]. For instance, $F_{ST}$s between East Asian populations and European ranged from 0.08 to 0.11 in this study, which was in agreement with $F_{ST}$ between Beijing Han and people of European ancestry (around 0.10) using genome-wide data. Besides, $F_{ST}$s between populations in different continents are around 0.15 in this study and the genome-wide study [43]. The PCA and admixture analysis are the most commonly used approaches to investigate genetic relationships between populations. They have been applied to multiple InDel datasets in previous studies [20,42,44]. Similar to the analysis of $F_{ST}$s, results of the PCA and admixture analysis showed significant genetic homogeneity between Hui and East Asian populations. Although the genetic imprint of East Asian ancestry in Hui was not in question, these two non-quantitative methods could not prove whether or not Hui received any genetic contribution from populations outside East Asia with limited InDel loci.

Therefore, we attempted to apply the three-population test that was systematically introduced by Patterson *et al*. [35] to this dataset. The f3 statistics of Hui were significantly negative and greater than those of Kazak and Uigur. The three-population test is robust to the sample size [43]. Additionally, this result was in accord with admixture signals observed in Kazak and Uigur. Thus, we first statistically detected the gene flow from both eastern and western Eurasians in Ningxia Hui, although the latter is limited.

Evidence from autosomal STR loci indicated that the spread of Islamic faith in India was predominantly cultural transformation associated with minor gene flow from West Asia [45]. Like what happened in India, the drastic decrease of admixture proportion of the western Eurasian populations in Ningxia Hui may also be caused by their long-term intermarriage with local residents during the course of Islamization. Thus, our results also support a cultural diffusion during the Islamization of Ningxia Hui. It is of great interest to know when and how these gene flow events from the West occurred, which might require much more information drawn from genome-wide analysis. Finally, since the Hui ethnic group in different regions of China may have complex and diverse sources [6,13], their population genetic behaviours could also vary.

Ethics. All of the donors had been adequately informed and signed the informed consent before sample collection. This study was approved by the ethics committee of Xi'an Jiaotong University Health Science Center and conducted in accordance with the human and ethical research principles of Xi'an Jiaotong University Health Science Center, China.

Data accessibility. The datasets supporting this article have been uploaded as part of the electronic supplementary material.

Authors' contributions. B.Z. and S.W. did part of the data processing and wrote the main manuscript text; H.S. wrote part of the main manuscript text and did the manuscript modification; H.Z. and R.S. designed the research and did the manuscript modification. All authors reviewed the manuscript.

Competing interests. We have no competing interests.

Funding. This project was supported by the National Natural Science Foundation of China (NSFC, grant nos. 81373248, 81525015 and 11371101) and Shaanxi Provincial Programs for Science and Technology Development (grant no. 2016SF-031).

Acknowledgements. The authors thank Yuchen Wang for running ADMIXTOOLS.

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
