## [Reviewer comments · Royal Society Open Science]

Review History

RSOS-190358.R0 (Original submission)

Review form: Reviewer 1

Is the manuscript scientifically sound in its present form?

No

Are the interpretations and conclusions justified by the results?

Yes

Is the language acceptable?

Yes

Is it clear how to access all supporting data?

Yes

Do you have any ethical concerns with this paper?

No

Have you any concerns about statistical analyses in this paper?

I do not feel qualified to assess the statistics

Recommendation?

Major revision is needed (please make suggestions in comments)

Comments to the Author(s)

The study aimed at a relevant long-standing controversial issue, the ethnic origin of Hui population in Ningxia. The authors tried to solve the problem with the genotyping results of a 30-InDel-loci panel in 129 unrelated individuals. The conclusion was that the Hui population mainly inherited genetic information from East Asian populations and that the West Eurasian populations' contribution were weak. Several methods were applied to achieve that conclusion, including F_{st} value, multidimensional scaling (MDS), principle component analysis (PCA) and the 3-population test.

However, I am afraid that the manuscript has not reached the publication level, and can be improved in following aspects:

1. As described in the part of Introduction, multiple studies have been carried out on this debate. And it is hard to solve such debate with limited markers and small sample size (as argued in reference 6 of this manuscript). The reason why a simple multiplexed kit of 30 InDel loci can solve it was not well discussed. Without such discussions, this manuscript is no more than adding some sterile arguments into this debate.
2. The authors admitted in the manuscript that these InDel loci were not AIMs (Page 3, Line50). With these markers applied on ancestry inference, a larger number of loci would be needed to achieve a roughly equivalent efficiency of AIMs. As we know, the F_{st} value applied in the manuscript can be an important parameter measuring the genetic distance between two populations. An F_{st} value around 0.15 indicates a moderate genetic differentiation between two population, which, to my knowledge, is much closer than the actual difference between Hui and Mexican populations. So, I am afraid that in this study, only 30 non-AIM InDel loci were not enough and the F_{st} values calculated based on these loci could hardly reflect the distance between populations accurately. And this should be discussed.
3. For biallelic markers, such as InDel loci and SNPs, a lower sample size than STR loci would be needed to achieve accurate allele frequencies in populations, which are critical for population genetics calculations, including F_{st} and 3-population tests. The sample size of 129 unrelated individuals is around the minimum requirement for population data obtainment of biallelic markers recommended by FSI: Genetics. A detailed discussion about the sufficiency of the sample size should be taken out.
4. The X-bars of Figure 4 are uncomfortably reversed. And there are some overlaps between the 95% confidence intervals of $f_3(A; B, \text{Hui})$ and $f_3(A; B, \text{Uigur})$ or $f_3(A; B, \text{Kazak})$ with the same A and B. Although the overlap of CIs does not represent the absence of significant difference, the differences between Hui and the other two mixture populations should not be expressed only by a single figure. Strict hypothesis testing and P-values are needed to confirm them.
5. As described in the section related of PCA results (Page 3, Line 25), the two components used as X and Y-bars in Figure 2 accounted for just 18.38% of the total variance, i.e. above 80% of the original information is missing through dimensionality reduction. Therefore, the ability of Figure 2 to explain the accurate genetic distance between populations is highly suspected, and should be discussed detailly.

6. Contents in parts of Materials and Methods, Results and Discussion were confusingly mixed up. For example, the details of the 3-population tests were first introduced in the part of Discussion, instead of Results like other methods.

Some parts need to be rewritten:

- a) The obtainment of the data in other populations should also be described in the part of Materials and Methods;
- b) Methods applied, including Fst values, MDS, PCA, admixture analysis and 3-population tests, as well as the meaning of important parameters in such methods, NOT just the software for each method, should be briefly introduced in the part of Materials and Methods.
- c) The results of each method applied should be listed and, if necessary, simply discussed in the part of Results.
- d) The interpretation of results and comparison with other studies should be mainly written in the part of Discussion. And the detailed discussions related to the questions I raised above should be added to this part.
- e) The part related to situation happens in India (Page 4, Line 16-19) should also be written in the part of Discussion.
- f) The parts of Results and Discussion can be written together if not necessary to be separated.

7. There are some confusions on the population names.

- a) Are the "Western Eurasian" populations in some parts of the manuscript the same as "European" populations in the figures and some other parts? If so, they should be named in the same way.
- b) Is the word "Uigur" the correct name of the population? I found two different ways of spelling, "Uygur" and "Uighur", in the references.
- c) Which one of the terms "Kazak" and "Kazakh" is correct?

8. There are some mistakes in references

- a) reference 30 was described as the origin source of the data of 30 InDel loci in one of the three Han populations, Tibetan, Kazak and Uigur, but what I found with the citation name was a study focus on 17 Y-STRs.
- b) References 43 and 6 are the same one.

9. Several language improvements can be applied:

- a) Page 2, Line 12, "Due to some advantages of short amplicon size" → "Due to the advantage of short amplicon size";
- b) Page 2, Line 14, "has attracted more and more attention" → "is attracting more and more attentions";
- c) Page 2, Line 53, "Hardy-Weinberg equilibrium test did not find significant deviation from expected value for all of the 30 InDel markers" → "No significant deviation from expected value on any of the 30 InDel markers was found via Hardy-Weinberg equilibrium test";
- d) Page 4, Line 1, "western Eurasians had weak genetic contribution to Ningxia Hui compared with Uigur and Kazak" → "western Eurasians had weak genetic contribution to Ningxia Hui, compared with their contributions to Uigur and Kazak".

Review form: Reviewer 2

Is the manuscript scientifically sound in its present form?

Yes

Are the interpretations and conclusions justified by the results?

Yes

Is the language acceptable?

Yes

Is it clear how to access all supporting data?

Yes

Do you have any ethical concerns with this paper?

No

Have you any concerns about statistical analyses in this paper?

No

Recommendation?

Major revision is needed (please make suggestions in comments)

Comments to the Author(s)

The authors made a population study to approve the INDEL panel (Qiagen DIPplex® Investigator kit) was a useful tool in forensic individual identification and population genetics analysis for Hui population. The proper analysis of the Genetic data suggested that the Ningxia

Hui had close genetic relationships with Chinese populations, with minor gene flow from western Eurasian populations. From a population study perspective, these data were valuable for the scientific community.

As for the content of this research paper, some aspects would benefit of revisions in order to improve its overall quality and usefulness for the scientific community:

1. The language should be improved. The text is understandable but A number of sentences are clumsy and poorly constructed. For example, line 34 "and their formation is due to the simple cultural diffusion rather than demic diffusion".
2. Clearer information is needed about definition, subdivision and recruitment criteria for the samples. How is Hui population defined and identified? Was this self-definition? Does it involve particular lifestyle/religion/customs? Are they linguistically distinct from other populations?
3. The references cited in this text were all before 2017. The authors should note that some population genetic analysis in Hui population has been carried out in recent years. In addition, some WestAsia Populations, such as Iraqi (Tomas, C., et al. "Thirty autosomal insertion-deletion polymorphisms analyzed using the DIPplex® Investigator kit in populations from Iraq, Lithuania, Slovenia, and Turkey."), Afghanistan and Pakistan(He, Guanglin, et al. "A comprehensive exploration of the genetic legacy and forensic features of Afghanistan and Pakistan Mongolian-descent Hazara."), should be considered in population genetic analysis part.
4. Xinjiang Hui population had been studied using this INDEL panel (Xie, Tong, et al. "A set of autosomal multiple InDel markers for forensic application and population genetic analysis in the Chinese Xinjiang Hui group.") . Is there any difference between these two Hui populations?
5. The analysis results were not fully discussed in the discussion section.
6. The labeling of the figure is inconsistent in Figure 1 . Please check similar situation within the whole manuscript!

7. Genotyping data for each individual has not uploaded.

Decision letter (RSOS-190358.R0)

08-Aug-2019

Dear Dr Zhou,

The editors assigned to your paper ("Genetic affinity between Ningxia Hui and eastern Asian populations revealed by a set of InDel Loci") have now received comments from reviewers.

Both reviewers raise a substantive number of points and criticisms that impact upon the rigor and reliability of your conclusions. These points will need careful thought and consideration.

We would like you to revise your paper in accordance with the referee suggestions which can be found below (not including confidential reports to the Editor). Please note this decision does not guarantee eventual acceptance.

Please submit a copy of your revised paper before 31-Aug-2019. Please note that the revision deadline will expire at 00.00am on this date. If we do not hear from you within this time then it will be assumed that the paper has been withdrawn. In exceptional circumstances, extensions may be possible if agreed with the Editorial Office in advance. We do not allow multiple rounds of revision so we urge you to make every effort to fully address all of the comments at this stage. If deemed necessary by the Editors, your manuscript will be sent back to one or more of the original reviewers for assessment. If the original reviewers are not available, we may invite new reviewers.

- Data accessibility

It is a condition of publication that all supporting data are made available either as supplementary information or preferably in a suitable permanent repository. The data

accessibility section should state where the article's supporting data can be accessed. This section should also include details, where possible of where to access other relevant research materials such as statistical tools, protocols, software etc can be accessed. If the data have been deposited in an external repository this section should list the database, accession number and link to the DOI for all data from the article that have been made publicly available. Data sets that have been deposited in an external repository and have a DOI should also be appropriately cited in the manuscript and included in the reference list.

If you wish to submit your supporting data or code to Dryad (<http://datadryad.org/>), or modify your current submission to dryad, please use the following link:
<http://datadryad.org/submit?journalID=RSOS&manu=RSOS-190358>

- **Competing interests**

- **Authors' contributions**

- **Acknowledgements**

- **Funding statement**

Kind regards,
Lianne Parkhouse
Editorial Coordinator
Royal Society Open Science
openscience@royalsociety.org

on behalf of Professor Joris Veltman (Associate Editor) and Steve Brown (Subject Editor)
openscience@royalsociety.org

Reviewers' Comments to Author:

Reviewer: 1

Comments to the Author(s)

The study aimed at a relevant long-standing controversial issue, the ethnic origin of Hui population in Ningxia. The authors tried to solve the problem with the genotyping results of a 30-InDel-loci panel in 129 unrelated individuals. The conclusion was that the Hui population mainly inherited genetic information from East Asian populations and that the West Eurasian populations' contribution were weak. Several methods were applied to achieve that conclusion, including F_{st} value, multidimensional scaling (MDS), principle component analysis (PCA) and the 3-population test.

However, I am afraid that the manuscript has not reached the publication level, and can be improved in following aspects:

1. As described in the part of Introduction, multiple studies have been carried out on this debate. And it is hard to solve such debate with limited markers and small sample size (as argued in reference 6 of this manuscript). The reason why a simple multiplexed kit of 30 InDel loci can solve it was not well discussed. Without such discussions, this manuscript is no more than adding some sterile arguments into this debate.

2. The authors admitted in the manuscript that these InDel loci were not AIMs (Page 3, Line50). With these markers applied on ancestry inference, a larger number of loci would be needed to achieve a roughly equivalent efficiency of AIMs. As we know, the F_{st} value applied in the manuscript can be an important parameter measuring the genetic distance between two populations. An F_{st} value around 0.15 indicates a moderate genetic differentiation between two population, which, to my knowledge, is much closer than the actual difference between Hui and Mexican populations. So, I am afraid that in this study, only 30 non-AIM InDel loci were not enough and the F_{st} values calculated based on these loci could hardly reflect the distance between populations accurately. And this should be discussed.

3. For biallelic markers, such as InDel loci and SNPs, a lower sample size than STR loci would be needed to achieve accurate allele frequencies in populations, which are critical for population genetics calculations, including F_{st} and 3-population tests. The sample size of 129 unrelated individuals is around the minimum requirement for population data obtainment of biallelic markers recommended by FSI: Genetics. A detailed discussion about the sufficiency of the sample size should be taken out.

4. The X-bars of Figure 4 are uncomfortably reversed. And there are some overlaps between the 95% confidence intervals of $f_3(A; B, \text{Hui})$ and $f_3(A; B, \text{Uigur})$ or $f_3(A; B, \text{Kazak})$ with the same A and B. Although the overlap of CIs does not represent the absence of significant difference, the differences between Hui and the other two mixture populations should not be expressed only by a single figure. Strict hypothesis testing and P-values are needed to confirm them.

5. As described in the section related of PCA results (Page 3, Line 25), the two components used as X and Y-bars in Figure 2 accounted for just 18.38% of the total variance, i.e. above 80% of the original information is missing through dimensionality reduction. Therefore, the ability of Figure 2 to explain the accurate genetic distance between populations is highly suspected, and should be discussed detailly.

6. Contents in parts of Materials and Methods, Results and Discussion were confusingly mixed

up. For example, the details of the 3-population tests were first introduced in the part of Discussion, instead of Results like other methods.

Some parts need to be rewritten:

- a) The obtainment of the data in other populations should also be described in the part of Materials and Methods;
- b) Methods applied, including Fst values, MDS, PCA, admixture analysis and 3-population tests, as well as the meaning of important parameters in such methods, NOT just the software for each method, should be briefly introduced in the part of Materials and Methods.
- c) The results of each method applied should be listed and, if necessary, simply discussed in the part of Results.
- d) The interpretation of results and comparison with other studies should be mainly written in the part of Discussion. And the detailed discussions related to the questions I raised above should be added to this part.
- e) The part related to situation happens in India (Page 4, Line 16-19) should also be written in the part of Discussion.
- f) The parts of Results and Discussion can be written together if not necessary to be separated.

7. There are some confusions on the population names.

- a) Are the "Western Eurasian" populations in some parts of the manuscript the same as "European" populations in the figures and some other parts? If so, they should be named in the same way.
- b) Is the word "Uigur" the correct name of the population? I found two different ways of spelling, "Uygur" and "Uighur", in the references.
- c) Which one of the terms "Kazak" and "Kazakh" is correct?

8. There are some mistakes in references

- a) reference 30 was described as the origin source of the data of 30 InDel loci in one of the three Han populations, Tibetan, Kazak and Uigur, but what I found with the citation name was a study focus on 17 Y-STRs.
- b) References 43 and 6 are the same one.

9. Several language improvements can be applied:

- a) Page 2, Line 12, "Due to some advantages of short amplicon size" → "Due to the advantage of short amplicon size";
- b) Page 2, Line 14, "has attracted more and more attention" → "is attracting more and more attentions";
- c) Page 2, Line 53, "Hardy-Weinberg equilibrium test did not find significant deviation from expected value for all of the 30 InDel markers" → "No significant deviation from expected value on any of the 30 InDel markers was found via Hardy-Weinberg equilibrium test";
- d) Page 4, Line 1, "western Eurasians had weak genetic contribution to Ningxia Hui compared with Uigur and Kazak" → "western Eurasians had weak genetic contribution to Ningxia Hui, compared with their contributions to Uigur and Kazak".

Reviewer: 2

Comments to the Author(s)

The authors made a population study to approve the INDEL panel (Qiagen DIPlex® Investigator kit) was a useful tool in forensic individual identification and population genetics analysis for Hui population. The proper analysis of the Genetic data suggested that the Ningxia

Hui had close genetic relationships with Chinese populations, with minor gene flow from

western Eurasian populations. From a population study perspective, these data were valuable for the scientific community.

As for the content of this research paper, some aspects would benefit of revisions in order to improve its overall quality and usefulness for the scientific community:

1. The language should be improved. The text is understandable but A number of sentences are clumsy and poorly constructed. For example, line 34 "and their formation is due to the simple cultural diffusion rather than demic diffusion".
2. Clearer information is needed about definition, subdivision and recruitment criteria for the samples. How is Hui population defined and identified? Was this self-definition? Does it involve particular lifestyle/religion/customs? Are they linguistically distinct from other populations?
3. The references cited in this text were all before 2017. The authors should note that some population genetic analysis in Hui population has been carried out in recent years. In addition, some WestAsia Populations, such as Iraqi (Tomas, C., et al. "Thirty autosomal insertion-deletion polymorphisms analyzed using the DIPplex® Investigator kit in populations from Iraq, Lithuania, Slovenia, and Turkey."), Afghanistan and Pakistan(He, Guanglin, et al. "A comprehensive exploration of the genetic legacy and forensic features of Afghanistan and Pakistan Mongolian-descent Hazara."), should be considered in population genetic analysis part.
4. Xinjiang Hui population had been studied using this INDEL panel (Xie, Tong, et al. "A set of autosomal multiple InDel markers for forensic application and population genetic analysis in the Chinese Xinjiang Hui group.") . Is there any difference between these two Hui populations?
5. The analysis results were not fully discussed in the discussion section.
6. The labeling of the figure is inconsistent in Figure 1 . Please check similar situation within the whole manuscript!
7. Genotyping data for each individual has not uploaded.

Author's Response to Decision Letter for (RSOS-190358.R0)

See Appendix A.

RSOS-190358.R1 (Revision)

Review form: Reviewer 1

Is the manuscript scientifically sound in its present form?

Yes

Are the interpretations and conclusions justified by the results?

Yes

Is the language acceptable?

Yes

Do you have any ethical concerns with this paper?

No

Have you any concerns about statistical analyses in this paper?

No

Recommendation?

Accept as is

Comments to the Author(s)

No comments

Review form: Reviewer 2

Is the manuscript scientifically sound in its present form?

Yes

Are the interpretations and conclusions justified by the results?

Yes

Is the language acceptable?

Yes

Do you have any ethical concerns with this paper?

No

Have you any concerns about statistical analyses in this paper?

No

Recommendation?

Accept as is

Comments to the Author(s)

The authors properly answered the proposed question. The manuscript should be accepted for publication.

Decision letter (RSOS-190358.R1)

29-Nov-2019

Dear Dr Zhou,

It is a pleasure to accept your manuscript entitled "Genetic affinity between Ningxia Hui and eastern Asian populations revealed by a set of InDel Loci" in its current form for publication in

Royal Society Open Science. The comments of the reviewer(s) who reviewed your manuscript are included at the foot of this letter.

on behalf of the Associate Editor, and Professor Steve Brown (Subject Editor)
openscience@royalsociety.org

Reviewer comments to Author:

Reviewer: 1
Comments to the Author(s)

No comments

Reviewer: 2
Comments to the Author(s)

The authors properly answered the proposed question. The manuscript should be accepted for publication.

Appendix A

We are grateful to two reviewers for their insightful comments. We have extensively revised the manuscript to fully address the comments. Our point-to-point responses are provided below.

Reviewer: 1

Comments to the Author(s)

The study aimed at a relevant long-standing controversial issue, the ethnic origin of Hui population in Ningxia. The authors tried to solve the problem with the genotyping results of a 30-InDel-loci panel in 129 unrelated individuals. The conclusion was that the Hui population mainly inherited genetic information from East Asian populations and that the West Eurasian populations' contribution were weak. Several methods were applied to achieve that conclusion, including F_{st} value, multidimensional scaling (MDS), principle component analysis (PCA) and the 3-population test.

However, I am afraid that the manuscript has not reached the publication level, and can be improved in following aspects:

1. As described in the part of Introduction, multiple studies have been carried out on this debate. And it is hard to solve such debate with limited markers and small sample size (as argued in reference 6 of this manuscript). The reason why a simple multiplexed kit of 30 InDel loci can solve it was not well discussed. Without such discussions, this manuscript is no more than adding some sterile arguments into this debate.

Response: Thanks for the comment. Our study mainly focuses on the forensic use of this InDel panel and the detection of admixture signal using some population genetic methods. We do not attempt to thoroughly solve this problem with such limited samples and loci. However, our result statistically supports the genetic contribution from western Eurasians to Hui, which proves the necessity of studying the origin of Hui using genome wide data. Additionally, we have rewritten the Discussion section as you suggested.

2. The authors admitted in the manuscript that these InDel loci were not AIMs (Page 3, Line50). With these markers applied on ancestry inference, a larger number of loci would be needed to achieve a roughly equivalent efficiency of AIMs.

As we know, the F_{st} value applied in the manuscript can be an important parameter measuring the genetic distance between two populations. An F_{st} value around 0.15 indicates a moderate genetic differentiation between two population, which, to my knowledge, is much closer than the actual difference between Hui and Mexican populations. So, I am afraid that in this study, only 30 non-AIM InDel loci were not enough and the F_{st} values calculated based on these loci could hardly reflect the distance between populations accurately. And this should be discussed.

Response: Thanks for the comment. The F_{st} value is influenced by many factors, such as genetic markers, estimation methods and population sizes. Indeed, F_{st} value around 0.15 is the borderline of moderate and great differentiation. However, our result (Supplementary Table 1) is generally in agreement with that from 1000 genomes data computed by Bhatia et al. (Bhatia G, Patterson N,

Sankararaman S, et al. Estimating and interpreting FST: the impact of rare variants[J]. Genome research, 2013, 23(9): 1514-1521.) In that article, Using three different F_{st} estimators (WC, Nei, and Hudson), F_{st} between CEU and CHB ranges from 0.106 to 0.112, and F_{st} between CHB and YRI ranges from 0.161 to 0.175. In our result, $F_{st}(\text{Han/Hui} : \text{European})$ is around 0.08 and $F_{st}(\text{Han/Hui} : \text{Mexican})$ ranges from 0.14 to 0.22. Although the number of InDel loci is limited, our result is rational and consistent with genome wide data. Additionally, we have added this discussion to the Discussion section as you suggested.

3. For biallelic markers, such as InDel loci and SNPs, a lower sample size than STR loci would be needed to achieve accurate allele frequencies in populations, which are critical for population genetics calculations, including F_{st} and 3-population tests. The sample size of 129 unrelated individuals is around the minimum requirement for population data obtainment of biallelic markers recommended by FSI: Genetics. A detailed discussion about the sufficiency of the sample size should be taken out.

Response: Thanks for the comment. We have added corresponding discussion in Discussion Section including the sufficiency of sample size.

4. The X-bars of Figure 4 are uncomfortably reversed. And there are some overlaps between the 95% confidence intervals of $f_3(A; B, \text{Hui})$ and $f_3(A; B, \text{Uigur})$ or $f_3(A; B, \text{Kazak})$ with the same A and B. Although the overlap of CIs does not represent the absence of significant difference, the differences between Hui and the other two mixture populations should not be expressed only by a single figure. Strict hypothesis testing and P-values are needed to confirm them.

Response: Thanks for the comment. We have reversed the X-bars of Figure 4. However, the software ADMIXTOOLS only provided Z-score that measure the standard error. It did not apply a strict hypothesis testing with P-value.

5. As described in the section related of PCA results (Page 3, Line 25), the two components used as X and Y-bars in Figure 2 accounted for just 18.38% of the total variance, i.e. above 80% of the original information is missing through dimensionality reduction. Therefore, the ability of Figure 2 to explain the accurate genetic distance between populations is highly suspected, and should be discussed detailly.

Response: Thanks for the comment. Although the first two components accounted for 18.38% of the total variance, relationships between populations that they reflected were the same as the rest of components. In addition, we have added this part of discussion in the revised manuscript.

6. Contents in parts of Materials and Methods, Results and Discussion were confusingly mixed up. For example, the details of the 3-population tests were first introduced in the part of Discussion, instead of Results like other methods.

Some parts need to be rewritten:

- a) The obtainment of the data in other populations should also be described in the part of Materials and Methods;*
- b) Methods applied, including Fst values, MDS, PCA, admixture analysis and 3-population tests, as well as the meaning of important parameters in such methods, NOT just the software for each method, should be briefly introduced in the part of Materials and Methods.*
- c) The results of each method applied should be listed and, if necessary, simply discussed in the part of Results.*
- d) The interpretation of results and comparison with other studies should be mainly written in the part of Discussion. And the detailed discussions related to the questions I raised above should be added to this part.*
- e) The part related to situation happens in India (Page 4, Line 16-19) should also be written in the part of Discussion.*
- f) The parts of Results and Discussion can be written together if not necessary to be separated.*

Response: Thanks for the comment. We have moved 3-population tests to the Results Section.

- a) We have described the obtainment of the data from other populations in the part of Materials and Methods.
- b) We have rewritten this part of Materials and Methods to briefly introduced these methods.
- c) We have listed results in Results section and discussed detailly in Discussion section.
- d) We have added the detailed discussions to Discussion.
- e) We have briefly introduced the Islamization of local residents in India in the revised manuscript.
- f) We have merged the Discussion and Conclusion section. But we still wrote Result and Discussion separately.

7. There are some confusions on the population names.

- a) Are the “Western Eurasian” populations in some parts of the manuscript the same as “European” populations in the figures and some other parts? If so, they should be named in the same way.*
- b) Is the word “Uigur” the correct name of the population? I found two different ways of spelling, “Uygur” and “Uighur”, in the references.*
- c) Which one of the terms “Kazak” and “Kazakh” is correct?*

Response: In this manuscript, “Western Eurasian” populations refers to “European” populations and “Western Asian” populations. Both “Uigur” and “Uygur”/“Uighur” are correct spelling of the same population, which is the same case with “Kazak”/“Kazakh”.

8. There are some mistakes in references

- a) reference 30 was described as the origin source of the data of 30 InDel loci in one of the three Han populations, Tibetan, Kazak and Uigur, but what I found with the citation name was a study focus on 17 Y-STRs.*

b) References 43 and 6 are the same one.

Response: Thanks for the comment. We have rectified these errors.

9. Several language improvements can be applied:

a) Page 2, Line 12, “Due to some advantages of short amplicon size” → “Due to the advantage of short amplicon size”;

b) Page 2, Line 14, “has attracted more and more attention” → “is attracting more and more attentions”;

c) Page 2, Line 53, “Hardy-Weinberg equilibrium test did not find significant deviation from expected value for all of the 30 InDel markers” → “No significant deviation from expected value on any of the 30 InDel markers was found via Hardy-Weinberg equilibrium test”;

d) Page 4, Line 1, “western Eurasians had weak genetic contribution to Ningxia Hui compared with Uigur and Kazak” → “western Eurasians had weak genetic contribution to Ningxia Hui, compared with their contributions to Uigur and Kazak”.

Response: Thanks for the suggestion. We have applied these language improvements in the revised manuscript.

Reviewer: 2

Comments to the Author(s)

The authors made a population study to approve the INDEL panel (Qiagen DIPplex® Investigator kit) was a useful tool in forensic individual identification and population genetics analysis for Hui population. The proper analysis of the Genetic data suggested that the Ningxia

Hui had close genetic relationships with Chinese populations, with minor gene flow from western Eurasian populations. From a population study perspective, these data were valuable for the scientific community.

As for the content of this research paper, some aspects would benefit of revisions in order to improve its overall quality and usefulness for the scientific community:

1. The language should be improved. The text is understandable but A number of sentences are clumsy and poorly constructed. For example, line 34 "and their formation is due to the simple cultural diffusion rather than demic diffusion".

Response: Thanks for the comment. We have improved the language of this manuscript thoroughly.

2. Clearer information is needed about definition, subdivision and recruitment criteria for the

samples. How is Hui population defined and identified? Was this self-definition? Does it involve particular lifestyle/religion/customs? Are they linguistically distinct from other populations?

Response: Thank you for your valuable advice. For clearer description of the sample collection part, modified sentences are provided in the revised manuscript, which are displayed as follows: We collected bloodstain samples of 129 unrelated healthy Hui individuals in the Ningxia Hui Autonomous Region, China according to their identity cards. No kindship existed among the samples within at least three generations, and no migration events happened in their family history as declared. All of the donors had been adequately informed and signed the informed consent before sample collection.

People of Hui origin can be found in most of the counties and cities throughout the country, especially in the Ningxia Hui Autonomous Region and Gansu, Qinghai, Henan, Hebei, Shandong and Yunnan provinces and the Xinjiang Uygur Autonomous Region. The name Hui is an abbreviation for "Huihui," which first appeared in the literature of the Northern Song Dynasty (960-1127). It referred to the Huihe people (the Uigurs) who lived in Anxi in the present-day Xinjiang and its vicinity since the Tang Dynasty (618-907).

The Hui people believe in Islamic and the Islamic religion had a deep influence on the life style of the Hui people. For instance, soon after birth, an infant was to be given a Huihui name by an ahung (imam); wedding ceremonies must be witnessed by ahungs; a deceased person must be cleaned with water, wrapped with white cloth and buried coffinless and promptly in the presence of an ahung who serves as the presider. Men were accustomed to wearing white or black brimless hats, especially during religious services, while women were seen with black, white or green scarves on their head, which is a habit derived from religious practices. The Huis never eat pork nor the blood of any animal or creature that died of itself, and they refuse to take alcohol. These taboos originated in the Koran of the Moslems. The Huis are very particular about sanitation and hygiene. Likewise, before attending religious services, they have to observe either a "minor cleaning," i.e. wash their face, mouth, nose, hands and feet, or a "major cleaning," which requires a thorough bath of the whole body.

The Huihuis initially used the Arab, Persian and Han languages. However, in the course of their long years living with the Hans, and especially due to the increasing number of Hans joining their ranks, they gradually spoke the Han language only, while maintaining certain Arab and Persian phrases. Huihui culture originally had been characterized by influences from the traditional culture of Western Asia and assimilation from the Han culture. However, due to the introduction of the Han language as a common language, the tendency to assimilate the Han culture became more obvious. The Huihuis began to wear clothing like the Hans. Huihui names were still used, but Han names and surnames became accepted and gradually became dominant.

3. The references cited in this text were all before 2017. The authors should note that some population genetic analysis in Hui population has been carried out in recent years. In addition, some WestAsia Populations, such as Iraqi (Tomas, C., et al. "Thirty autosomal insertion-deletion polymorphisms analyzed using the DIPplex® Investigator kit in populations from Iraq, Lithuania, Slovenia, and Turkey."), Afghanistan and Pakistan(He, Guanglin, et al. "A comprehensive exploration of the genetic legacy and forensic features of Afghanistan and Pakistan Mongolian-

descent Hazara."), should be considered in population genetic analysis part.

Response: Thanks for the comment. It will be a very interesting research subject to explore the genetic relationships between Hui and some Western Asian populations. However, according to our results, the genetic contribution to Ningxia Hui from western Eurasians is limited. In addition, we compared the genetic component of Hui with those of Kazak and Uigur which are typical admixed populations of the East and the West. Therefore, taking these populations into consideration will not change the conclusion.

4. Xinjiang Hui population had been studied using this INDEL panel (Xie, Tong, et al. "A set of autosomal multiple InDel markers for forensic application and population genetic analysis in the Chinese Xinjiang Hui group."). Is there any difference between these two Hui populations?

Response: Thanks for the comment. According to the F_{st} and MDS results, these two Hui populations are similar since they both have the closest genetic relationship with Han Chinese. But they still have slight differences in allele frequencies. Due to the complexity of Hui people in different regions of China, it will be a significant and valuable work to explore the origin of Hui with more samples and more genetic markers in the future. In this article, we only focused on the forensic application of 30 InDels in Ningxia Hui and the detection of admixture signal using some population genetic methods.

5. The analysis results were not fully discussed in the discussion section.

Response: Thanks for the comment. We have moved 3-population tests to the Results Section and rewritten the Discussion Section. We have added discussions of comparisons with other studies and the significance of our contribution as suggested by you and another reviewer.

6. The labeling of the figure is inconsistent in Figure 1. Please check similar situation within the whole manuscript!

Response: Thanks for the comment. We have checked these figures. We guessed that this problem may be caused by the low resolution of Figure 1 or the overlap of "GuangdongHan" and "ShanghaiHan". We have changed legends of some populations in Figure 1 to make it clearer.

7. Genotyping data for each individual has not uploaded.

Response: Thanks for the comment. The genotyping data has been uploaded as Supplementary Material.